# Changes in sea-surface temperature and atmospheric circulation patterns associated with reductions in Arctic sea-ice cover in recent decades

Lejiang Yu[1], Shiyuan Zhong[2]

5    [1]SOA Key Laboratory for Polar Science, Polar Research Institute of China, Shanghai, China

[2]Department of Geography, Environment and Spatial Sciences, Michigan State University, East Lansing, MI, USA

Correspondence to:    Dr. Shiyuan Zhong (zhongs@msu.edu)

**Abstract.** In recent decades, the Arctic sea ice has been declining at a rapid pace as the Arctic is warmed at a rate of twice the global average. The underlying physical mechanisms for the Arctic warming and accelerated sea ice retreat are not fully understood. In this study, we apply a relatively novel statistical method called Self-Organizing Maps (SOM) to examine the trend and variability of autumn Arctic sea ice in the past three decades and their relationships to large-scale atmospheric circulation changes. Our statistical results show that the anomalous autumn Arctic Dipole (AD) (Node 1) and the Arctic Oscillation (AO) (Node 9) could explain in a statistical sense as much as 50% of autumn sea ice decline between 1979 and 2016. The Arctic atmospheric circulation anomalies associated with anomalous sea surface temperature patterns over the North Pacific and North Atlantic influence Arctic sea ice primarily through anomalous temperature and water vapour advection and associated radiative feedback.

**1 Introduction**

In recent decades, the Arctic sea ice has been decreasing at an unprecedented rate (Rothrock et al., 1999; Parkinson, 2014). The accelerated retreat in Arctic sea ice exerts a significant impact not only on the marine and terrestrial ecological systems of the Arctic (Post et al., 2013), but also on the environment of the mid latitude (Mori et al., 2014; Kug et al., 2015).

The underline mechanisms for the Arctic sea ice decline remain to be a subject of active research. Studies have suggested that both anthropogenic forcing due to greenhouse gas and aerosol emissions (Min et al., 2008; Notz and Marotzke, 2012; Gagné et al., 2015) and natural mechanisms at a wide range of scales contribute to the observed Arctic sea ice decline. Local processes, including surface thermal inversion (Bintanja et al., 2011), atmospheric lapse-rate (Pithan and Mauritsen, 2014), ice-albedo feedback (Flanner et al., 2011) and water vapour and cloud radiative feedback (Sedlar et al., 2011), have been found to affect Arctic sea ice. On the other end, global sea-surface temperature (SST) and pressure oscillations, such as the Arctic Dipole (AD) (Wang et al., 2009), the Atlantic Multidecadal Oscillation (AMO) (Park and Latif, 2009), the Arctic Oscillation (AO) (Rigor et al., 2002; Deser and Teng, 2008), the North Atlantic Oscillation (NAO) (Koenigk et al., 2009), and the Pacific Decadal Oscillation (PDO) (Ding et al., 2014), have also been linked to the Arctic sea ice variations and the recent declining trend. Understanding the relative contributions from these multi-scale processes to Arctic sea ice decline is vital not only for forecasting sea ice conditions but also for projecting climate change and its impact on the Arctic environment and beyond (Stroeve et al., 2007; Day et al., 2012; Swart et al., 2015; Ding et al.,

2017).

In this study, we examine the contributions of changes in large-scale atmosphere and ocean circulations to the trends in Arctic sea ice by applying the Self Organizing Maps (SOM) method (Kohonen, 2001). A relatively new neural network-based method, SOM is superior to some other feature-extracting or clustering methods in that it describes the continuum of atmospheric and oceanic states with a manageable number of representative patterns as compared to only providing useful information on aggregate patterns by Empirical Orthogonal Function (EOF) and other similar methods. Over the past decade, SOM has been widely used in atmosphere and ocean sciences (Hewitson and Crane, 2002; Leloup et al., 2007; Johnson et al., 2008; Lee et al., 2011; Chu et al., 2012). For example, SOM was used to explain the eastward shift of NAO since the late 1970s (Johnson et al., 2008). The SOM approach was also used to examine the contributions of different El Niño Southern Oscillation (ENSO) to the SST trend in the tropical Pacific Ocean (Johnson, 2013).

We apply SOM to a monthly sea ice concentration dataset for the period 1979 - 2016. We will show how much the recent declining trend in the Arctic sea ice concentrations may be associated with the low frequency atmospheric circulation modes related to SST anomalies over the Pacific and Atlantic Oceans. Although the analyses have been carried out for all four seasons, we will show the results for autumn only since Arctic sea ice reduction in autumn has strongest influence on the wintertime atmospheric circulations over Eurasia and North America (Francis et al., 2009; Petoukhov and Semenov, 2010; Peings and Magnusdottir, 2014).

**2 Methods**

The period of this study is from 1979-2016. The Arctic sea ice anomaly analysis utilizes the monthly sea ice concentration dataset produced by the United States National Snow and Ice Data Centre (NSIDC) (http://nsidc.org/data/NSIDC-0051). The NSIDC data are on a horizontal grid of 25 km $\times$25 km with a polar stereographic projection. The corresponding SST patterns are analyzed using the $1\,°$latitude $\times 1\,°$ longitude SST data from the Hadley Centre (Rayner et al., 2003) (http://www.metoffice.gov.uk/hadobs/hadisst/). The SST anomaly patterns are linked to well-known modes of climate variability characterized by several climate indices including the Pacific Decadal Oscillation (PDO) (Mantua et al., 1997) defined as the first EOF mode of SST anomalies in the North Pacific Ocean, the Atlantic Multidecadal Oscillation (AMO) (Enfield et al., 2001) defined as the

10-year running mean of detrended SST anomalies in the Atlantic Ocean north of the equator, and the Arctic Oscillation (AO) and Arctic Dipole (AD) defined as the first and second EOF modes of the 1000-hPa geopotential height anomalies north of 70$^{\circ}$N (Wang et al., 2009).

The atmospheric circulations are determined using data from the 1.5 $^{\circ}$latitude $\times$1.5 $^{\circ}$longitude European Centre for Medium-Range Weather Forecasts (ECMWF) Reanalysis (ERA-Interim) (Dee et al., 2011). Serreze et al. (2012) assessed humidity data from ERA-Interim reanalyses and found only small bias of less than 8% at 1000 hPa for boreal autumn and smaller bias above 1000 hPa. Ding et al. (2017) showed that the ERA-Interim reanalyses can represent reliably the observed circulation, radiation flux, temperature, and water vapour. The surface and upper-air atmospheric variables examined in this study are all extracted from ERA-Interim, except for outgoing long-wave radiation (OLR) that is obtained from the gridded monthly OLR data produced by the US National Oceanic and Atmospheric Administration (NOAA) (http://www.esrl.noaa.gov/psd/data/gridded/data.interp_OLR.html) (Liebmann and Smith, 1996).

The SOM technique is utilized to extract patterns of Arctic sea ice concentrations. A neural network-based method, SOM uses unsupervised learning to determine generalized patterns in complex data. The technique can reduce multidimensional data into two-dimensional array consisting of a matrix of nodes. Each node in the array has a reference vector that displays a spatial pattern of the input data. All patterns in the two-dimensional array represent the full continuum of states in the input data. The SOM algorithm also is a clustering technique, but unlike other clustering technique, it does not need a priori decisions on data distribution. Unlike the EOF analysis, the SOM technique does not require the orthogonality of two spatial patterns. A detailed description of the SOM algorithm can be found in Kohonen (2001).

In this study, the SOM technique is used to categorize anomalous seasonal sea ice concentration patterns north of 50 $^{\circ}$N for autumn (September-October-November). Seasonal anomalies are calculated by subtracting the climatology for the season relative to the study period 1979-2016 from the seasonal means. The anomalous sea ice pattern for each autumn is assigned to the best-matching SOM pattern on the basis of minimum Euclidean distance, a measure of straight-line distance between two points in Euclidean space and calculated by the product of root-mean-square error and the square root of the number of grid points. Spatial correlations between sea ice field for each autumn and its corresponding

best-matching SOM pattern are calculated and the overall mean spatial correlation coefficients obtained by averaging over the 38 autumns are used to determine the number of SOM nodes or grids to be used in the current analysis (Lee and Feldstein, 2013). Table 1 shows the overall mean spatial correlation coefficients for several SOM grid configurations with nodes ranging from 2×2 to 4×5. As expected, the overall correlation increases as the number of nodes increases. The largest jump between any two grid configurations occurs from a 2×4 grid to a 3×3 grid. Thus, a 3×3 grid is selected for the SOM analysis. While a higher spatial correlation can be achieved with larger number of grids, the 3x3 grid is able to capture the main variability pattern in the autumn Arctic sea ice at the sacrifice of some details that can be depicted by larger number of grids. The frequency of occurrence for each SOM node is defined as the number of autumns that node represents divided by the total number of autumns (38) in the study period. The contribution of each SOM pattern to the trends in the Arctic sea ice concentrations is calculated by the product of each SOM pattern and a rate determined by temporal linear regression of the number of the projections of each pattern for each autumn (Johnson, 2013). The sum of these contributions presents the trends in Arctic sea ice explained in the statistical sense by the SOM patterns, which in this case indicates trends resulting from low-frequency variability. The significance of the trends in the time series for each SOM pattern is tested using the Student's t-test. Residual trends are calculated by subtracting SOM-explained trends from the total trends.

**3 SOM results**

Sea ice concentration anomalies occur mainly in the marginal seas from the Barents Sea to the Beaufort Sea, with maximum anomalies in the Barents Sea and the Kara Sea, and over the Beaufort Sea and the East Siberian Sea (Fig. 1). Nodes 1, 2 and 4 depict all negative anomalies, whereas Nodes 5, 6, 8 and 9 exhibit all positive anomalies. Nodes 3 and 7, on the other hand, show a mixed pattern with opposite changes in the Barents Sea in the Atlantic sector of the Arctic Ocean and the Beaufort Sea and East Siberian Sea over the Pacific sector. The largest change with similar strength over the Pacific and the Atlantic sectors are depicted by Nodes 1 and 9, and both nodes also have the highest (24%) frequency of occurrences. The other nodes that are either all positive or all negative exhibit larger spatial variability in the strengths of the signals with much stronger signal in either the Atlantic sector (Nodes 2, 5 and 8) or the Pacific sector (Nodes 4 and 6). The two mixed patterns show similar strength over the two sectors, but the pattern with positive anomalies over the Beaufort Sea and the East Siberian Sea and negative

anomalies over the Barents Sea, as depicted by Node 3, occurs much more frequently (13%) than the opposite pattern represented by Node 7 (8%).

The trends in the frequency of occurrence for each SOM pattern and their contribution to the trends in the Arctic autumn sea ice concentrations are examined and the results are shown in Fig. 2. The nodes with spatially uniform changes appear to be separated into two clusters with those showing all positive anomalies (Nodes 5, 6, 8 and 9) appearing in the 80s and 90s and those having all negative anomalies (Nodes 1 and 4) appearing after 2000. The transition from all positive anomalies in the earlier part, to all negative anomalies in the later part, of the time series is consistent with the trends in the observed Arctic sea ice concentration during the same time period. Not surprisingly, the transition appears to be dominated by the two strongest and most frequent patterns denoted by Node 9 (all positive and occurring from the 1980s through mid 1990s) and Node 1 (all negative and occurring exclusively after 2005). Only these two nodes have linear trends that are statistically significant at above 95% confidence level. The slopes of the trend lines for these two nodes are opposite but the values are similar (0.027 $yr^{-1}$ for Node 1 and -0.021 $yr^{-1}$ for Node 9). The other nodes have statistically insignificant trends with magnitudes less than 0.01 $yr^{-1}$.

The spatial patterns of the autumn sea ice concentration trends explained in a statistical sense by each node and by all nodes are shown in Figs. 3 and 4. Together, the nine SOM nodes explain statistically about 60% of total trends in the autumn sea ice loss (Fig. 4b). Among them Node 1 explains statistically the largest portion (33%) of the total trend, followed by Node 9 (21%), with the other 7 nodes together accounting for only 6% of the loss.

For comparison, the SOM patterns and their occurrence time series for a larger grid (3x5) are provided in Supplement (Figs. S1 and S2). As expected, with more nodes, the 3x5 grid depict more details and each node has smaller frequency compared to the 3x3 grid. However, the dominant nodes (also nodes 1 and 9) show nearly identical patterns as those in the 3x3 grid. Like the 3×3 grid, nodes 1 and 9 in the 3×5 grid make greater contributions to the trend in autumn Arctic sea ice than other nodes (Fig. S3). Also as expected, the trend explained by nodes 1 and 9 in the 3x5 grid is smaller (46%) compared to 54% by the same two nodes in the 3×3 grid.

**4 Potential Mechanisms**

To explain statistically the spatial patterns depicted by the two dominant nodes, we made composite maps over the years of occurrences for Nodes 1 and 9, respectively, and Figs. 5-7 show composite patterns of the anomalous SST and OLR (as a proxy for tropical convection) and anomalous atmospheric circulations represented by the 500-hPa geopotential height, 850-hPa wind, surface-750-hPa specific humidity, surface downward longwave radiation, and surface air temperature.

For Node 1, the SST composite pattern resembles negative phase of the PDO in the North Pacific and positive phase of the AMO in the North Atlantic (Fig. 5). The positive SST anomalies over the tropical western Pacific produce anomalous local convection, exciting a wave train (Hoskins and Karoly, 1981; Jin and Hoskins, 1995; Higgins and Mo, 1997), which propagates northeastwards to North Pacific, North America, North Atlantic, and Eurasia (Fig. 5). In mid-latitudes, the anomalous SST and the nodes

of the wave train interacts to produce atmospheric circulation anomalies. Over the Arctic, the SST and height patterns are similar to those associated with a negative phase AD, with a negative 500-hPa height anomaly center over Greenland and the Baffin Bay, and a positive anomaly center over the Kara Sea. The zonal pressure gradient between the two centers induces anomalous low-troposphere southwesterly and southerly winds over the North Atlantic Ocean (Fig. 6) that transport warm and moist air from North

Atlantic into the Arctic Ocean north of Eurasia, thus increasing surface air temperature and humidity and reducing sea ice concentration in the Arctic (Fig. 7). The higher moisture content in the Arctic surface air also facilitates the occurrence of water vapour and cloud radiative feedback process (Sedlar et al., 2011) during which increased downward longwave radiation (Fig. 7) enhances surface warming and sea ice melting. The anomalous high pressure produces subsidence and the adiabatic warming associated with

the sinking air contributes to the sea ice loss (Ding et al., 2017) (Not shown). The opposite may occur in the region under the anomalous low pressure center. But the warm southwesterly winds over the northeastern Canada also favor the sea ice loss there. Gong et al. (2017) and Lee et al. (2017) showed that the increasing downward longwave radiation related to the changes of mid-latitude atmospheric circulation leads to the wintertime Arctic warming. Our focus here is on autumn sea ice trend and we

use the low-level water vapour field that is found to resemble more closely to the surface downward radiation fields compared to the total column water (not shown) used in the two studies. Like their studies, the North Atlantic is the main passage of water vapour into the Arctic. But over the Bering Sea

and Europe no water vapour is transported into the Arctic in our studies. Their study period ranges from 1980 to 2010, which is different from our study period (1979-2016).

For Node 9, the SST composite is characterized by negative anomalies in both the Pacific and the Atlantic except for areas of the tropical Pacific and the west coast of North America (Fig.5). The negative SST anomalies over the tropical western Pacific, corresponding to suppressed convective activities, generate one wave train that is in opposite phase to the wave train associated with Node 1. Similarly, the nodes of the wave train interact with the local SST anomalies over the mid-latitude regions, producing a nearly opposite (but not symmetrical) anomalous atmospheric circulation pattern to that of Node 1. Over the Arctic, the patterns resemble a positive phase AO (Fig.5). Anomalous northeasterly winds in the North Atlantic Ocean induced by anomalous Icelandic low are unfavourable for warm air intrusion into the Arctic Ocean (Fig. 6), a result also indicated in a previous study (Kim et al., 2017). Despite the occurrences of anomalous southwesterly winds over the Barents Sea and southeasterly winds over the Sea of Okhotsk, the cold advection from anomalous cold sea water prevents the Arctic sea ice from melting. Meanwhile the cold advection also reduces water vapor content in the lower troposphere and the resulting smaller downward longwave radiation facilitates the occurrence of negative surface air temperature and positive Arctic sea ice anomalies north of Eurasia (Fig. 7). The northerly winds over the Beaufort Sea, northeastern Canada and Greenland Sea also contribute to the decrease in surface air temperature and increase in the sea ice concentration.

The opposite patterns in Nodes 1 and 9 can be largely associated with the differences in the water vapor-radiation feedback process resulting from anomalous temperature and especially water vapour transport by anomalous atmospheric circulations associated with different patterns of SST anomalies over the North Pacific and Atlantic. The patterns of SST anomalies are nearly symmetric for the two nodes over the North Atlantic, but they are somewhat asymmetric over the North Pacific. For Node 1, there is one center of high SST over the central North Pacific, whereas Node 9 is associated with two centers of low SST: one over the Coast of Japan and another in central North Pacific. These differences in the SST anomaly patterns lead to different wave trains and high-latitude atmospheric circulations.

For the composite analysis often used in previous literature, lower (higher) Arctic sea concentration years refer to after (before) 2000.

The SOM-based composites in the above discussion, which are made using years indicated (Fig. 2) by the occurrences for Node 1 (2007-2013, 2015-2016), and, separately, for node 9 (1980-1982,

1986-1989, 1992, 1996), are compared to composites of years of high (before 2000) and low (after 2000) Arctic sea ice coverage according to the Autumn Arctic sea ice time series (Figure 8). The composited SST patterns over the North Pacific are almost a mirror image to each other, indicating a positive (negative) phase PDO before (after) 2000 (Fig. 8). On the other hand, the composited SST patterns corresponding to Node 1 and Node 9 are not symmetrical (Fig. 5). Similar situations occur over North Atlantic. The magnitude and significant level of SOM-based SST composites in Fig. 5 are also higher than the time-series-based composites in Fig. 8. Similarly, in mid-latitudes, the composites of the anomalous 500hPa geopotential heights before and after 2000 are nearly symmetrical (Fig. 8) and the significant level and amplitude of the wave train are also lower than those in Fig. 5. At high latitudes, the anomalous atmospheric circulations in Fig. 8 show a mixed pattern of AO and AD and have no extreme centers. However, Fig. 5 exhibits clear AO and AD patterns and extreme centers, revealing more clearly the relationship between the anomalous sea ice concentration and anomalous atmospheric circulations. Based on these examples, the SOM-based composites allows for better depiction of atmospheric and SST conditions corresponding to sea ice anomaly patterns compared to typical time-series-based composite approach.

One drawback of the SOM-based composite is the relatively small number of composite members (Node 1 and Node 9 each has 9 members), which makes the composite more vulnerable to the influence of extreme cases. To examine this issue, the standard deviations of the SOM-based composites are compared to those of time-series-based composites that have much larger members (17 members for after-2000- composites and 21 members for before-2000-composites) and the results for the anomalous SST and 500-hPa height are shown in Figs. S4 and S5. The spatial patterns and magnitudes of the standard deviations are similar between the Node-1 and after-2000 composites and between Node-9 and before-2000 composites. The F-test indicates that except for small patches there are no statistically significant differences in the standard deviations between the small member SOM-based composites and the large-member typical composites (Fig. S6), suggesting it is unlikely the SOM-based composites are strongly influenced by extreme cases as a result of small composite members.

**5 Conclusions**

We investigate the potential mechanisms for the autumn arctic sea ice decline for the period 1979-2016 using the SOM method. Our results show that more than half of the autumn Arctic sea ice loss may be associated with the changes in the temperature and water vapour transport and the associated water vapour radiation feedback resulting from anomalous atmospheric circulations linked to SST anomalies over the North Pacific and North Atlantic. This result provides further evidence that the mid-latitude SST anomalies play a vital role in the accelerated Arctic sea ice decline in recent decades (Ding et al., 2017; Yu et al., 2017). The main result is that the opposite pattern of the Arctic sea ice anomalies during the early (positive) and later (negative) parts of the 1979-2016 period is not associated with opposite phase of an atmospheric circulation mode, but instead the change can be largely associated with two different atmospheric circulation patterns (AO and AD) associated with an asymmetry in the anomalous SST distributions over the North Pacific. The teleconnections between the Arctic sea ice variability and mid-latitude SST anomalies suggest that on a decadal or longer time scale it may be necessary to include the Arctic sea ice and mid-latitude SST interactions or feedbacks in any investigations of Arctic warming and sea ice decline and their potential influence on mid-latitude weather and climate, an area of active research in recent years (Barnes and Screen, 2015; Overland and Wang, 2015; Francis and Skific, 2015). Finally, the results here help advance the knowledge about the relatively large contributions from the decadal-scale SST variability to Arctic climate change, though further studies using coupled global atmosphere-ocean-sea ice models are warranted to fully understand the physical mechanisms.

**Data Availability**

All data used in the current study are publicly available. The monthly sea ice concentration data are available from the National Snow and Ice Data Center (NSIDC) (http://nsidc.org/data/NSIDC-0051), the ERA-Interim reanalysis data are available from the European Center for Mid-Range Weather Forecasting (https://www.ecmwf.int/en/forecasts/datasets/reanalysis-datasets/era-interim) and the SST data are available from the Hadley Centre for Climate Prediction and Research (http://www.metoffice.gov.uk/hadobs/hadisst/).

**Competing interests**

The authors declare that they have no conflict of interest.

**Author Contributions**

L. Yu designed the study, with input from S. Zhong, and carried out the analyses. L. Yu and S. Zhong
prepared the manuscript.

**Acknowledgements**

This study is supported by National Key R&D Program of China (No.2017YFE0111700).

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

**Table 1. The 38-year (1979-2016) average of spatial correlations between the seasonal Arctic sea ice concentration and the corresponding SOM pattern for each year.**

| SOM grids | 2×2 | 2×3 | 2×4 | 3×3 | 3×4 | 3×5 | 4×4 | 4×5 |
|---|---|---|---|---|---|---|---|---|
| Correlation coefficients | 0.59 | 0.56 | 0.59 | 0.64 | 0.66 | 0.69 | 0.68 | 0.70 |

**Figure captions**

Fig.1. The SOM patterns of autumn (September-November) Arctic sea ice concentration for a $3 \times 3$ grid for the 1979-2016 period. The percentages at the top left of each panel indicate the frequency of occurrence of the pattern.

Fig. 2. Occurrence time series for each SOM pattern in Fig. 1.

Fig. 3. In the statistical sense, trends in the autumn Arctic sea ice concentration explained by each SOM pattern (Units: $yr^{-1}$).

Fig. 4. Total (a), SOM-explained (b) and residual (c) autumn Arctic sea ice concentration trends (Units: $yr^{-1}$). Dots in (a) indicate above 90% confidence level.

Fig. 5. Composites of anomalous sea surface temperature ($\mathrm{°C}$) and 500-hPa geopotential height (gpm) for nodes 1 and 9. Dotted regions denote above 90% confidence level.

Fig. 6. The same as Fig. 5, but for 850-hPa wind field. Shaded regions denote above 90% confidence level.

Fig. 7. The same as Fig. 5, but for integrated atmospheric water vapor from surface to 750 hPa (g $kg^{-1}$), accumulated surface downward longwave radiation (W $m^{-2} \times 10^{5}$) and surface air temperature ($\mathrm{°C}$). Dotted regions denote above 90% confidence level.

Fig. 8. Composites of anomalous sea surface temperature ($\mathrm{°C}$) and 500-hPa geopotential height (gpm) for after and before 2000. Dotted regions denote above 90% confidence level.

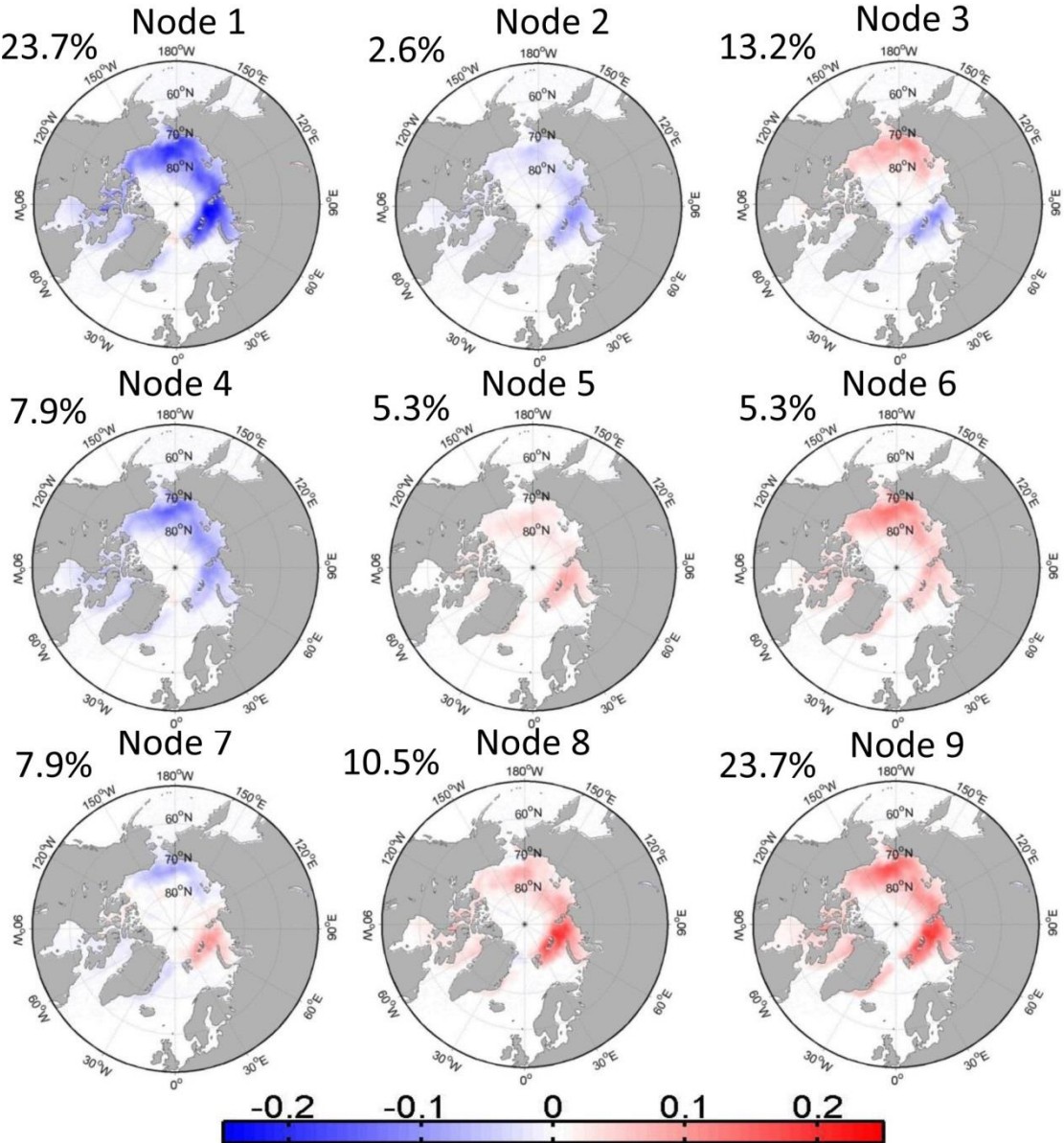

Figure1 The SOM patterns of the autumn (September-November) Arctic sea ice concentration for a 3×3 grid for the 1979-2016 period. The percentages at the top left of each panel indicate the frequency of occurrence of the pattern.

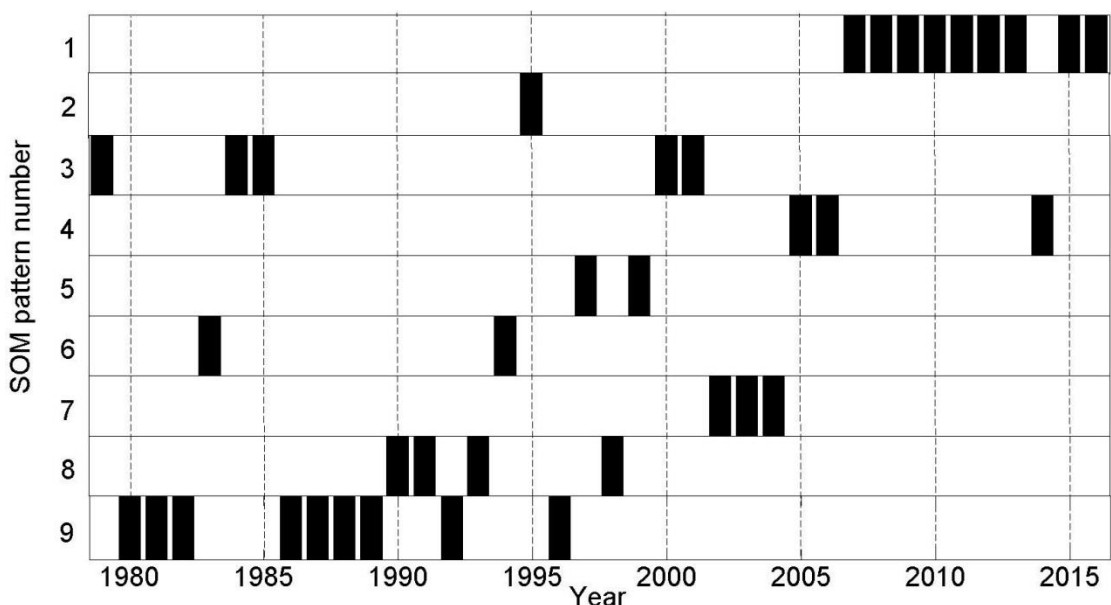

Figure 2. Occurrence time series for each SOM pattern in Figure 1.

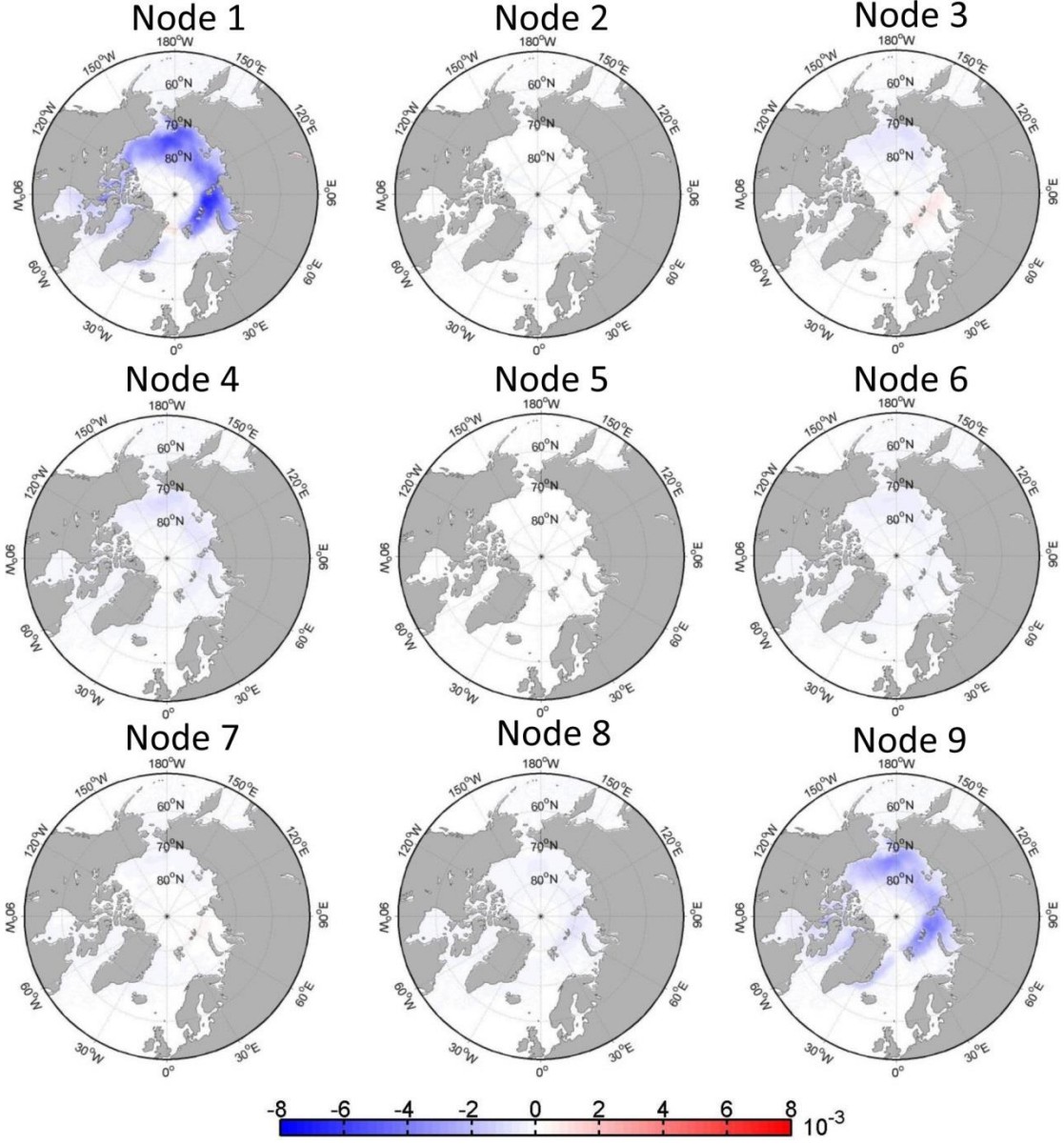

Figure 3. In the statistical sense, trends in the autumn Arctic sea ice concentration explained by each SOM pattern (Units: $yr^{-1}$).

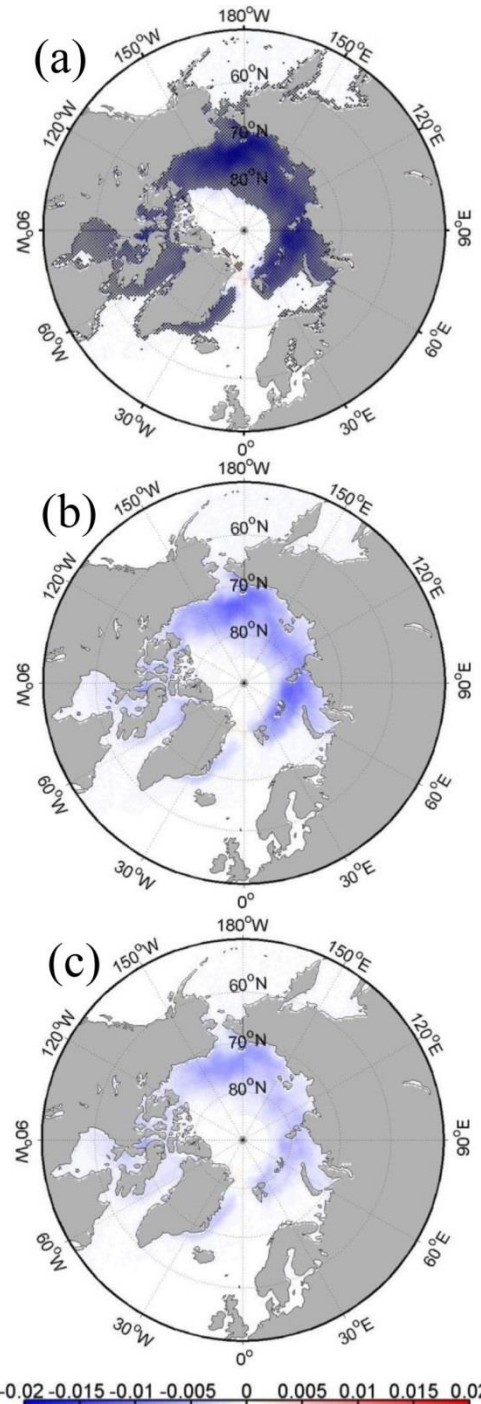

Figure 4. Total (a), SOM-explained (b) and residual (c) autumn Arctic sea ice concentration trends. (Units: yr$^{-1}$). Dots in (a) indicate above 95% confidence level.

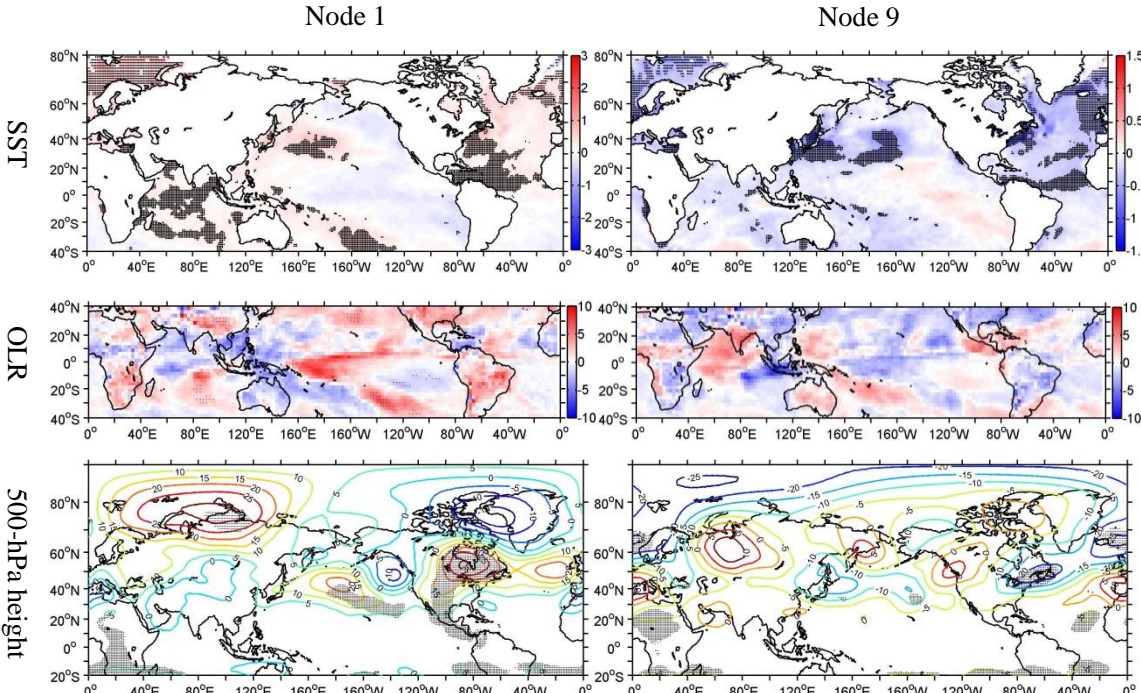

Figure 5. Composites of anomalous sea surface temperature (℃), outgoing long wave radiation (OLR) (W m$^{-2}$) and 500-hPa geopotential height (gpm) for nodes 1 and 9. Dotted regions denote above 90% confidence level.

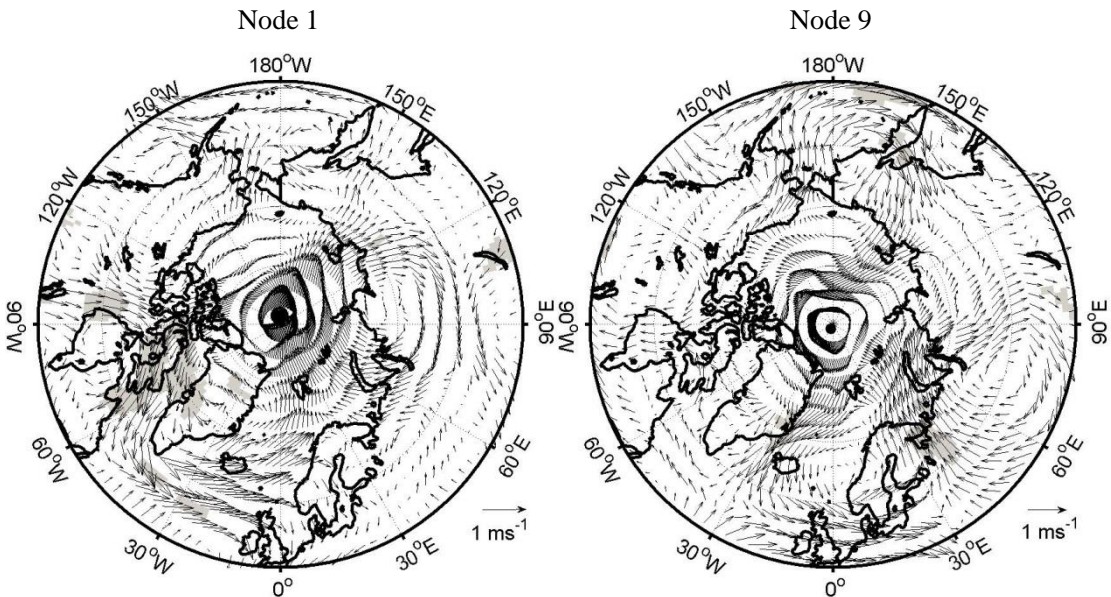

Figure 6. The same as Figure 5, but for 850 hPa wind field. Shaded regions denote above 90% confidence level.

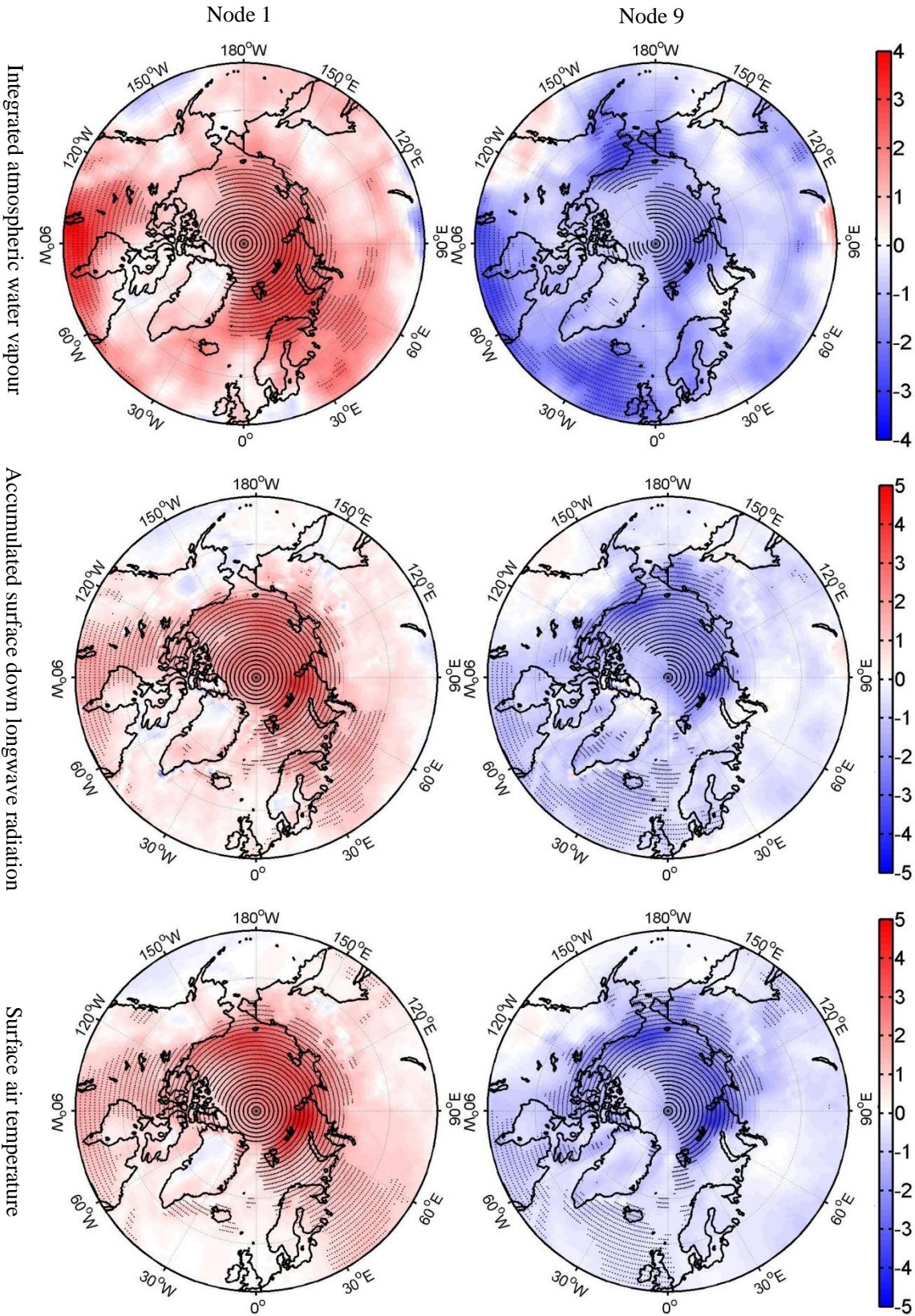

Figure 7. The same as Figure 5, but for integrated atmospheric water vapor from surface to 750 hPa (g kg$^{-1}$), accumulated surface downward longwave radiation (10$^5$Wm$^{-2}$), and surface air temperature (°C). Dotted regions denote above 90% confidence level.

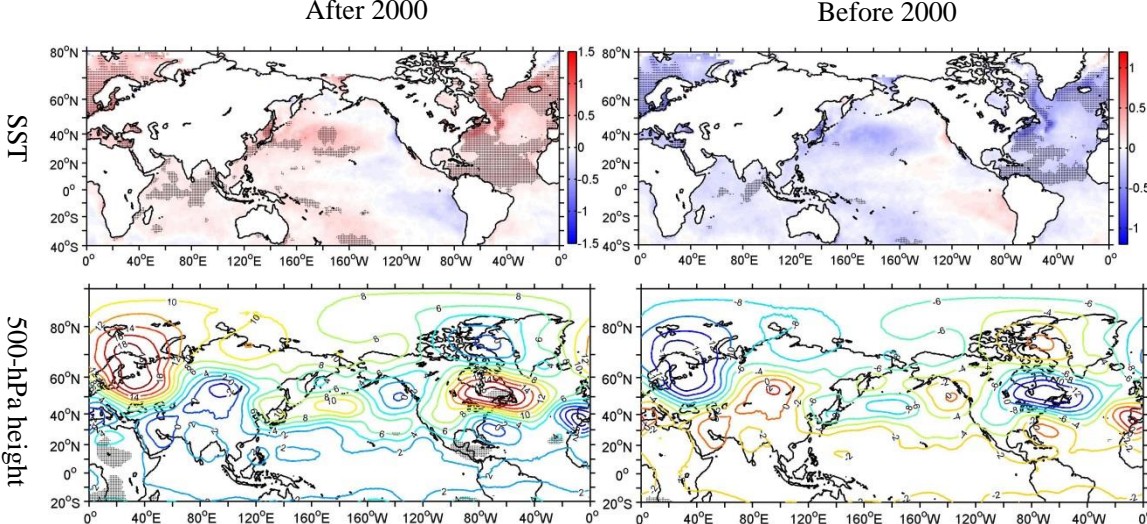

Figure 8. Composites of anomalous sea surface temperature (℃) and 500-hPa geopotential height (gpm) for after and before 2000. Dotted regions denote above 90% confidence level.