# Peer review of "Changes in sea-surface temperature and atmospheric circulation patterns associated with reductions in Arctic sea-ice cover in recent decades"

_Atmospheric Chemistry and Physics, 2018_

## Referee Comment (RC1) · Anonymous Referee #1 · 27 Apr 2018

General comments:

The authors have applied a self organising maps (SOM) algorithm to gridded Arctic sea ice concentration anomaly data for Sep-Oct-Nov 1979-2016 – a quantity that has a strong negative trend in this period – to obtain 9 spatial patterns (nodes). Nodes 1 and 9 have similar spatial distribution but opposite sign: node 9 (positive anomalies) is prominent early in the analysed period, and node 1 (negative anomalies) late in the period, and these two nodes account for much of the observed negative trend. Composites of sea surface temperature anomalies and several atmospheric variables are made using the 9 years for which node 1 is most prominent, and likewise for mode 9

(also 9 years). Features in the node 1 (node 9) composites are consistent with processes that reduce (increase) sea ice. It is claimed that the SOM-based composites provide a better depiction of patterns that influence sea ice trends, but there is no comparison with other representations to justify this claim. It is stated that the results 'help highlight the large contribution from the decadal-scale natural climate variability to Arctic climate change', but this is not clear from the results presented. On the evidence presented, the SOM approach does not seem to provide new insight. Further investigation is needed to demonstrate that the use of SOM is advantageous.

Specific comments:

Representation of sea ice concentration anomalies with 9 SOM spatial patterns (nodes, provided in Fig. 1) was selected. It is stated that results are similar with larger numbers of nodes, while smaller numbers are less representative. It is not clear how the spatial correlation coefficient (Table 1) was calculated however, and a change from 0.59 (2x4 nodes) to 0.64 (3x3 nodes) does not seem 'large' as stated on line 84.

For each of the 38 seasons available the best-matching node is tabulated (Fig 2). The 'frequency of occurrence' is defined as the number of times a pattern is thus selected, divided by 38: thus both node 1 and node 9 have a frequency of occurrence of 23.7% (9/38) in Fig. 1.

Nodes 1 and 9 have similar spatial distribution but opposite sign: node 9 (positive anomalies) is prominent early in the analysed period, and node 1 (negative anomalies) late in the period.

Fig. 3 shows trends associated with each node: this seems to be the node spatial pattern multiplied by a rate. (It is not clear how the rate is determined: possibly temporal linear regression of the projections of each pattern each season?) Nodes 1 and 9 are the main contributors. Fig. 4 illustrates how much of the total observed trend is associated with the SOM nodes. (It is not clear how this is calculated, but the text states about 60% in all is associated with the selected SOM nodes.)

Composites of sea surface temperature anomalies and various atmospheric quantities (anomalies from ERA-interim: 500hPa geopotential height, 850hPa wind, surface air T, surface downward longwave radiation, surface-to-750hPa water vapour) are made using years indicated in Fig. 2 for node 1 (2007-2013, 2015-2016), and, separately, for node 9 (1980-1982, 1986-1989, 1992, 1996). Effectively the SOM analysis provides the basis for these composites, which are illustrated in Figs. 5-7.

The composites are likely quite similar to composites of years when autumnal Arctic sea ice coverage was high versus low according to various other criteria: this should be discussed. The authors claim SOM allows 'better depiction of atmospheric circulation patterns that have significant impact on sea ice trends' (line 175), but no evidence is provided to support this claim, and this is a major weakness of this article.

With a relatively small number of cases (9) in each composite, some discussion of whether the composites are dominated by a few 'extremes' should be provided.

The analysis is largely descriptive. Various features in the composites are noted that are consistent with Arctic changes: e.g. for node 1 there are influences that favour sea ice reduction. Although suggestive in appearance, it is not evident that the SST anomalies and geopotential height anomalies in Fig. 5 are related as described. A zonal wavenumber 2 wavetrain (lines 134, 152) is not obvious.

Regarding downward longwave radiation and water vapour, how reliable are the ERA analyses in the Arctic?

While the analyses demonstrate associated changes in sea ice and in SST and atmospheric circulation, they do not in themselves seem to indicate cause and effect, so it is difficult to draw conclusions regarding mechanisms. The 'important finding' relating sea ice changes to asymmetry in North Pacific SST anomalies (lines 184-188) is not well justified. The claim of 'large contributions from the decadal-scale natural climate variability to Arctic climate change' (lines 193-194) does not seem well justified.

Other suggestions for technical corrections:

The term 'explained' is often used, but in the sense of statistical rather than physical explanation, which should be made clear.

The acronyms PDO, AMO, AD, AO are used without definition.

The title is rather misleading: the article is more about 'SST and atmospheric patterns associated with reductions in sea ice cover in recent decades'

---

## Author Comment (AC1) · 24 May 2018

We greatly appreciate the insightful reviews by the reviewers. Below are our responses (in red) to each of the review comments (in black)

Response to Reviewer 1

Specific comments: Representation of sea ice concentration anomalies with 9 SOM spatial patterns (nodes, provided in Fig. 1) was selected. It is stated that results are similar with larger numbers of nodes, while smaller numbers are less representative. It is not clear how the spatial correlation coefficient (Table 1) was calculated however, and

a change from 0.59 (2x4 nodes) to 0.64 (3x3 nodes) does not seem 'large' as stated on line 84. The spatial correlation values in Table 1 are obtained by two steps. First, for each autumn, spatial correlation between the sea ice anomaly pattern and the best matching SOM pattern is calculated (Lee and Feldstein, 2013). Second, the spatial correlations for all 38 autumns are averaged to obtain the mean spatial correlations shown in Table 1. These steps are repeated for the 8 pre-determined SOM grids in Table 1. The change from 0.59 to 0.64 is not large, but it is the largest gain between any two grids among the 8 grids tested. The choice of the grid is, admittedly, subjective. Obviously, higher spatial correlation can be achieved with larger grids. The 3x3 grid is able to capture the main variability pattern at the sacrifice of some details that can be depicted by larger grids. We have added clarification about the values in Table 1 and further justification of the choice of the grid.

For each of the 38 seasons available the best-matching node is tabulated (Fig 2). The'frequency of occurrence' is defined as the number of times a pattern is thus selected, divided by 38: thus both node 1 and node 9 have a frequency of occurrence of 23.7% (9/38) in Fig. 1.Nodes 1 and 9 have similar spatial distribution but opposite sign: node 9 (positive anomalies) is prominent early in the analysed period, and node 1 (negative anomalies) late in the period. Fig. 3 shows trends associated with each node: this seems to be the node spatial pattern multiplied by a rate. (It is not clear how the rate is determined: possibly temporal linear regression of the projections of each pattern each season?) Nodes 1 and 9 are the main contributors. The reviewers are correct about how the frequencies of occurrences and the trends or the rates of the change are determined. We have added clarification for these calculations Fig. 4 illustrates how much of the total observed trend is associated with the SOM nodes. (It is not clear how this is calculated, but the text states about 60% in all is associated with the selected SOM nodes.) The contribution of each SOM pattern to trends in Arctic sea ice concentration is calculated by the product of each SOM pattern and the rate, defined as temporal linear regression of the number of the projections of each pattern for each autumn. The sum of contributions from each node presents the total

observed trend associated with the SOM nodes. The 60% is estimated by the ratios of residual trends to the total trends that range from 10 to 90% across grid points with statistically significant trends. This has been clarified in the manuscript Composites of sea surface temperature anomalies and various atmospheric quantities (anomalies from ERA-interim: 500hPa geopotential height, 850hPa wind, surface air T, surface downward longwave radiation, surface-to-750hPa water vapour) are made using years indicated in Fig. 2 for node 1 (2007-2013, 2015-2016), and, separately, for node 9 (1980-1982, 1986-1989, 1992, 1996). Effectively the SOM analysis provides the basis for these composites, which are illustrated in Figs. 5-7. The composites are likely quite similar to composites of years when autumnal Arctic sea ice coverage was high versus low according to various other criteria: this should be discussed. The authors claim SOM allows 'better depiction of atmospheric circulation patterns that have significant impact on sea ice trends' (line 175), but no evidence is provided to support this claim, and this is a major weakness of this article. The following paragraph and a new figure (Figure 8) have been added that compares SOM-based composites with typical time-series based composite approach. The SOM-based composites in the above discussion, which are made using years indicated (Fig. 2) by the occurrences for Node 1 (2007-2013, 2015-2016), and, separately, for node 9 (1980-1982, 1986-1989, 1992, 1996), are compared to composites of years of high (before 2000) and low (after 2000) Arctic sea ice coverage according to the Autumn Arctic sea ice time series (Figure 8). The composited SST patterns over the North Pacific are almost a mirror image to each other, indicating a positive (negative) phase PDO before (after) 2000 (Fig. 8). On the other hand, the composited SST patterns corresponding to Node 1 and Node 9 are not symmetrical (Fig. 5). Similar situations occur over North Atlantic. The magnitude and significant level of SOM-based SST composites in Fig. 5 are also higher than the time-series-based composites in Fig. 8. Similarly, in mid-latitudes, the composites of the anomalous 500hPa geopotential heights before and after 2000 are nearly symmetrical (Fig. 8) and the significant level and amplitude of the wave train are also lower than those in Fig. 5. At high latitudes, the anomalous atmospheric circulations

in Fig. 8 show a mixed pattern of AO and AD and have no extreme centers. However, Fig. 5 exhibits clear AO and AD patterns and extreme centers, revealing more clearly the relationship between the anomalous sea ice concentration and anomalous atmospheric circulations. Based on these examples, the SOM-based composites allows for better depiction of atmospheric and SST conditions corresponding to sea ice anomaly patterns compared to typical time-series-based composite approach.

With a relatively small number of cases (9) in each composite, some discussion of whether the composites are dominated by a few 'extremes' should be provided. The following paragraph, along with three new figures, has been added. One drawback of the SOM-based composite is the relatively small number of composite members (Node 1 and Node 9 each has 9 members), which makes the composite more vulnerable to the influence of extreme cases. To examine this issue, the standard deviations of the SOM-based composites are compared to those of time-series-based composites that have much larger members (17 members for after-2000- composites and 21 members for before-2000-composites) and the results for the anomalous SST and 500-hPa height are shown in Figs. S1 and S2. The spatial patterns and magnitudes of the standard deviations are similar between the Node-1 and after-2000 composites and between Node-9 and before-2000 composites. The F-test indicates that except for small patches there are no statistically significant differences in the standard deviations between the small member SOM-based composites and the large-member typical composites (Fig.3), suggesting it is unlikely the SOM-based composites are strongly influenced by extreme cases as a result of small composite members.

The analysis is largely descriptive. Various features in the composites are noted that are consistent with Arctic changes: e.g. for node 1 there are influences that favour sea ice reduction. Although suggestive in appearance, it is not evident that the SST anomalies and geopotential height anomalies in Fig. 5 are related as described. A zonal wavenumber 2 wavetrain (lines 134, 152) is not obvious. We have added OLR composite analysis to determine the source of the wave train and further explain the

ACPD

relationship between SST and geopotential height anomalies. The anomalous convection activity over the tropical western Pacific Ocean can excite a wave train that propagates northeastwards. Over the mid-latitude North Pacific, the different local interaction between anomalous SST and wave train for Nodes 1 and 9 leads to different and asymmetrical pattern in the high latitudes.

Regarding downward longwave radiation and water vapour, how reliable are the ERA analyses in the Arctic? Ding et al. (2017) noted that the ERA-Interim analysis can represent reliably the observed circulation, radiation flux, temperature, and water vapour. Serreze et al. (2012) assessed humidity data from ERA-Interim analysis and found small bias of less than 8% at 1000 hPa for boreal autumn and smaller bias above 1000 hPa. Hence we considered ERA-Interim can represent reasonably well the observed downward longwave radiation and water vapour.

While the analyses demonstrate associated changes in sea ice and in SST and atmospheric circulation, they do not in themselves seem to indicate cause and effect, so it is difficult to draw conclusions regarding mechanisms. The 'important finding' relating sea ice changes to asymmetry in North Pacific SST anomalies (lines 184-188) is not well justified. The claim of 'large contributions from the decadal-scale natural climate variability to Arctic climate change' (lines 193-194) does not seem well justified. We added OLR analysis to show the source of the wave train. Local and asymmetrical interactions of SST and the node of wave train produce anomalous atmospheric circulations over the mid-high latitudes, which are related to anomalous Arctic sea ice concentrations. We agree that this and other analyses, which reveal the associations, do not demonstrate cause and effect. This is the major limitation of these types of climate diagnoses. We have softened the languages, e.g. important finding to major results, to reflect this point.

Other suggestions for technical corrections:

The term 'explained' is often used, but in the sense of statistical rather than physical

explanation, which should be made clear. We added 'in the statistical sense' in the sentence where 'explained' is used, or replaced the word 'explained' by 'is shown to be related' The acronyms PDO, AMO, AD, AO are used without definition. The definitions of PDO, AMO, AD, and AO were added in Dataset and Methods section The title is rather misleading: the article is more about 'SST and atmospheric patterns associated with reductions in sea ice cover in recent decades' The title is changed to 'The changes in sea-surface temperature and atmospheric circulation patterns associated with reductions in Arctic sea ice cover in recent decades'

Please also note the supplement to this comment:
https://www.atmos-chem-phys-discuss.net/acp-2018-127/acp-2018-127-AC1-supplement.pdf
* * *
[Figure]

Figure S1 The standard deviations of composite SST for Nodes 1 and 9, and after and before 2000.

**Fig. 1.**

[Figure]

Figure S2 The standard deviations of composite 500-hPa height for Nodes 1 and 9, and after and before 2000.

**Fig. 2.**

[Figure]

Figure S3 The significant test of standard deviation for SST and 500-hPa height between Node 1 and after 2000, and Node 9 and before 2000 using F test. Red dots denote above 95% confidence level.

**Fig. 3.**

---

## Referee Comment (RC2) · Anonymous Referee #2 · 19 Jun 2018

The authors used the Self Organizing Maps (SOM) method to examine both the variability and trend of autumn Arctic sea ice over the past few decades. They found that about 60% of the recent autumn Arctic sea ice decline can be explained by 9 intrinsic modes, and specifically, SST anomalies over the North Pacific and North Atlantic, resulting atmospheric circulation and water vapor radiative processes.

The application of the SOM method to Arctic sea ice looks new to me and some interesting results are found. However, I have some major comments about this manuscript. I would recommend publication in ACP when they are addressed.

Major comments:

1. About the number of nodes selected. Although the authors claimed that they chose 3x3 SOM grid because there is a large increase in correlation from 2x4 to 3x3, I feel the correlation increase looks pretty gradual to me and thus the choice of 3x3 is not very convincing. As the authors also claimed that "larger grids, ..., do not alter the results and conclusions", I would suggest the authors include this information perhaps in the Supplementary Materials to better support the conclusions. Since, based on Table 1, increase of nodes after 3x5 does not seem to increase the correlation anymore, I would suggest the authors provide the results using 3x5.

2. About reference of previous studies. Although the application of the SOM method to Arctic sea ice is new, some of the results and conclusions drawn from the analysis have been found in previous studies but the authors failed to include them. Some of the relevant studies are listed below -

Gong, T., S. Feldstein, and S. Lee, 2017: The Role of Downward Infrared Radiation in the Recent Arctic Winter Warming Trend. J. Climate, 30, 4937–4949, https://doi.org/10.1175/JCLI-D-16-0180.1

Lee, S., Gong , T., Feldstein, S. B.,Screen, J. A., & Simmonds, I. (2017).Revisiting the cause of the 1989– 2009Arctic surface warming using the sur-face energy budget: Downward infraredradiation dominates the surface fluxes.Geophysical Research Letters, 44,10,654–10,661. https://doi.org/10.1002/2017GL075375

I would suggest the authors cite these references and add discussions on the consistency/inconsistency as compared to previous studies.

Minor comments:

Line 36: "natural processes"? why are these processes all natural?

Lines 40-45: The authors failed to explain the advantages of the SOM method here. The EOF method also provides "a manageable number of representative patterns".

Line 48: NAO has been defined before

Line 49: ENSO has not been defined

Line 80: The authors might want to better define what is "Euclidean distance" in the SOM method

Line 85: smaller number of grids

Line 86: larger number of grids

Lines 96-98: The authors should reference Fig. 4a here?

Line 134 and 152: It's not easy to tell that it is zonal wave number 2 here.

Line 193: "decadal-scale natural climate variability to Arctic climate change", why the authors concluded natural here? Can't the SOM nodes include anthropogenic components too?

Fig. 4: I don't see dots in (a)?

---

## Author Comment (AC2) · 25 Jun 2018

Anonymous Referee #2 The authors used the Self Organizing Maps (SOM) method to examine both the variability and trend of autumn Arctic sea ice over the past few decades. They found that about 60% of the recent autumn Arctic sea ice decline can be explained by 9 intrinsic modes, and specifically, SST anomalies over the North Pacific and North Atlantic, resulting atmospheric circulation and water vapor radiative processes. The application of the SOM method to Arctic sea ice looks new to me and some interesting results are found. However, I have some major comments about this manuscript. I would recommend publication

in ACP when they are addressed. Major comments: 1. About the number of nodes selected. Although the authors claimed that they chose 3x3 SOM grid because there is a large increase in correlation from 2x4 to 3x3, I feel the correlation increase looks pretty gradual to me and thus the choice of 3x3 is not very convincing. As the authors also claimed that "larger grids, : : :, do not alter the results and conclusions", I would suggest the authors include this information perhaps in the Supplementary Materials to better support the conclusions. Since, based on Table 1, increase of nodes after 3x5 does not seem to increase the correlation anymore, I would suggest the authors provide the results using 3x5. As suggested by the reviewer, we now provide, as Supplemental Materials, the SOM patterns and their occurrence time series for the $3\times5$ grid (Figures S1and S2). As expected, with more nodes, the 3x5 grid depict more details and each node has smaller frequency compared to the 3x3 grid. However, the dominant nodes (also nodes 1 and 9) show nearly identical patterns as those in the 3x3 grid. Like the $3\times3$ grid, nodes 1 and 9 in the $3\times5$ grid make greater contributions to the trend in autumn Arctic sea ice than other nodes (Figure S3). Also as expected, the trend explained by nodes 1 and 9 in the 3x5 grid is smaller (46%) compared to 54% by the same two nodes in the $3\times3$ grid.

2. About reference of previous studies. Although the application of the SOM method to Arctic sea ice is new, some of the results and conclusions drawn from the analysis have been found in previous studies but the authors failed to include them. Some of the relevant studies are listed below - Gong, T., S. Feldstein, and S. Lee, 2017: The Role of Downward Infrared Radiation in the Recent Arctic Winter Warming Trend. J. Climate, 30, 4937–4949, https://doi.org/10.1175/JCLI-D-16-0180.1 Lee, S., Gong , T., Feldstein, S. B.,Screen, J. A., & Simmonds, I. (2017).Revisiting the cause of the 1989–2009Arctic surface warming using the sur-face energy budget: Downward infraredradiation dominates the surface ïnËĞC′ uxes.Geophysical Research Letters, 44,10,654–10,661. https://doi.org/10.1002/2017GL075375 I would suggest the authors cite these references and add discussions on the consistency/ inconsistency as compared to previous studies. Thanks for pointing to these references. We have now added them to

section 4 along with some discussion.

Minor comments: Line 36: "natural processes"? why are these processes all natural? We deleted 'natural'. Lines 40-45: The authors failed to explain the advantages of the SOM method here. The EOF method also provides "a manageable number of representative patterns". In the third paragraph of the Methods section, we provide the advantages of the SOM over EOF. Line 48: NAO has been defined before Changed Line 49: ENSO has not been defined Added Line 80: The authors might want to better define what is "Euclidean distance" in the SOM method The definition of Euclidean distance has been added. Line 85: smaller number of grids Changed Line 86: larger number of grids Changed Lines 96-98: The authors should reference Fig. 4a here? The sentence refers to Figure 1. Line 134 and 152: It's not easy to tell that it is zonal wave number 2 here. Changed Line 193: "decadal-scale natural climate variability to Arctic climate change", why the authors concluded natural here? Can't the SOM nodes include anthropogenic components too? 'natural climate variability' has been changed into 'SST variability'. Fig. 4: I don't see dots in (a)? Dots are enlarged.

Please also note the supplement to this comment:
https://www.atmos-chem-phys-discuss.net/acp-2018-127/acp-2018-127-AC2-supplement.pdf
* * *
[Figure]

**Node1** 21.1%

**Node2** 0.0%

**Node3** 13.2%

**Node4** 2.7%

**Node5** 0.0%

**Node6** 7.9%

**Node7** 2.6%

**Node8** 0.0%

**Node9** 18.4%

**Node10** 5.3%

**Node11** 2.6%

**Node12** 2.6%

**Node13** 5.3%

**Node14** 10.5%

**Node15** 7.9%

-0.2  -0.1  0  0.1  0.2

**Fig. 1.** Figure S1 The SOM patterns of the anomalous autumn (September-November) Arctic sea ice

[Figure]

**Fig. 2.** Figure S2. Occurrence time series for each SOM pattern in Figure S1.

Node1  Node2  Node3  Node4  Node5

Node6  Node7  Node8  Node9  Node10

Node11  Node12  Node13  Node14  Node15

-8  -6  -4  -2  0  2  4  6  8 $10^{-3}$

**Fig. 3.** Figure S3. Trends in the anomalous autumn Arctic sea ice concentration explained by each SOM pattern (Units: yr-1).

**After 2000**

**Before 2000**

**Node 1**

**Node 9**

**Fig. 4.** Figure S4 The standard deviations of composite SST for Nodes 1 and 9, and after and before 2000.

**After 2000**

**Before 2000**

**Node 1**

**Node 9**

**Fig. 5.** Figure S5 The standard deviations of composite 500-hPa height for Nodes 1 and 9, and after and before 2000.

Node 1 and after 2000

Node 9 and before 2000

SST

500-hPa height

**Fig. 6.** Figure S6 The significant test of standard deviation for SST and 500-hPa height between Node 1 and after 2000, and Node 9 and before 2000 using F test. Red dots denote above 95% confidence level.

**Supplement:**

[revised manuscript text omitted]

---

## Referee Report (RR1)

Originally 'Contributions from intrinsic low-frequency climate variability to the accelerated decline in Arctic sea ice in recent decades'

Title changed to 'Changes in sea surface temperature and atmospheric circulation patterns associated with reductions in Arctic sea-ice cover in recent decades

Referee report:

General comments:

The authors have provided more information and revised the discussions in response to the referees' comments, and have improved their manuscript.

Some aspects of the SOM patterns are still not clear to me, and the discussion of mechanisms needs more detail to be clearer.
I am not convinced that asymmetries in North Pacific SST anomalies are as important as the authors claim, so that aspect could be more speculative.

Thus I recommend further revision as follows:

Specific comments:

Are composite results sensitive to the selection of the SOM matrix of patterns? As in the example in the supplement, patterns similar to node1 and node 9 also occur and dominate other choices of matrix, and the selection of years for composites is likewise similar for other array choices.

On reading again the section on potential mechanisms, I think it could be made clearer and improved by providing more precise information about geographic regions mentioned, and linking more clearly to the sea ice concentration features in the nodes.

e.g. line 162 (and 187): specify latitude/longitude for the region of interest in the western tropical Pacific; likewise line 168 for the 'region between two centers', line 171 for region of reduced sea ice concentration, line 174 for the region of anomalous high pressure, line 176 for the region of anomalous low pressure, etc.

Line 207/208: apart from opposite sign, the differences in SST patterns are rather small, and it is difficult to attribute the differences in high-latitude circulation to the SST patterns without more direct evidence, so this should be stated as a hypothesis. (The SST anomaly sign difference alone and nonlinear effects might be the main cause of any SST-related circulation differences.)

It would be informative if one or two example scatterplots could be added to illustrate how particular regional values in ice cover and in atmospheric variables are related in in years making up composites.

Further minor comments:

It would be helpful to mention the use of composites in the abstract.

line 25: '...remain a subject of ..'

line 31: '...On larger scales, ...'

line 50: '... (ENSO) events to the ...'

line 78: what is the resolution of the OLR data grid?

Line 81: it would be helpful to add a statement about what is optimised in SOM

Line 85: I am not sure what the sentence 'All patterns in …' means, as the patterns represent a subset of the input data.

Line 94: it would be clearer to say 'The best matching SOM pattern for each autumn is determined on the basis of minimum Euclidean distance ….'. It would be useful to provide an equation for this. It would also be better to place this later in the paragraph (e.g. after line 105), after defining the patterns.

Line 107: It would be clearer to define frequency of occurrence as the number of autumns for which a pattern is selected as 'best matching' divided by the total number of autumns.

Line 110: I am not sure what is meant by 'linear regression of the number of the projections' : an equation would be useful.

Fig. 1: are the patterns normalised?
(It might be worth noting that the sign of the pattern is important, unlike EOFs for which the sign is arbitrary. An EOF analysis would give one pattern like nodes 1 and 9, with different signs of projection in early/late years in the data.)

Line 122: it would help to say that 24% corresponds to 9/38.

Line 127: omit 'much' (the occurrence difference is between 5/38 and 3/38)

Line 130: … Figs 2 and 3

Fig. 4: are the labels correct? The SOM-associated trend seems larger than the observed trend. Also, the 'dots' indicating significance are very hard to see in the figure.

Line 155: Suggest better to put 'To explore the processes associated with the spatial patterns …'

Line 213: better to also state the years used in the before and after 2000 composites.

Line 217: to me, the dominant feature of the SST patterns associated with node1 and node9 is their similarity, with small differences. Further evidence is required if circulation and sea-ice concentration differences are to be attributed to the influence of the small SST pattern differences – perhaps you could state that further

investigation is required to support your statements, particularly as this effect forms a large part of your conclusions in section 8.

Fig. 2 caption: Better to say that the figure indicates which mode best matched the observed sea ice concentration in each autumn.

Fig. 3 caption: Rather than 'explained', this figure just shows the linear trend associated with each pattern when projected on the observations?

Fig. 4: check the content, labelling and dots.

Fig. 5 and Fig. 8: It would be helpful to repeat in the caption the actual years used for each composite.

---

## Author Response (AR2)

l26: omit 'to be'

Change made

l53: 'much' > 'much of'

Change made

l80: 'by subtracting the climatology for the season relative to the period 1979-2016 from the seasonal means' — presumably you mean 'by subtracting the mean over a chosen season for each year from the mean for that season over all the years 1979-2016'?

It's the opposite. Subtracting B from A (A-B) where A is the mean for a chose season for each year and B is the mean for that season over all years in 1979-2016. Sentence changed to "subtracting the mean over a chosen season for each year from the mean for that season over all the years in 1979-2016"

l119: You have now defined 'frequency' but have not given explicit details of how you calculated 'frequency trends'. Presumably you calculate the time series of frequency for each individual year and then calculate the trend from that time series? Please confirm in the text.

Clarified

l152: 'vorciticity' > 'vorticity'

The word 'vorciticity' was a typo in the initial submission, which was corrected in the revised version.

[revised manuscript text omitted]

95    season for each year. The anomalous sea ice pattern for each autumn is assigned to the best-matching

SOM pattern on the basis of minimum Euclidean distance, a measure of straight-line distance between

two points in Euclidean space and calculated by the product of root-mean-square error and the square

root of the number of grid points. Spatial correlations between sea ice field for each autumn and its corresponding best-matching SOM pattern are calculated and the overall mean spatial correlation coefficients obtained by averaging over the 38 autumns are used to determine the number of SOM nodes or grids to be used in the current analysis (Lee and Feldstein, 2013). Table 1 shows the overall mean spatial correlation coefficients for several SOM grid configurations with nodes ranging from 2×2 to 4×5. As expected, the overall correlation increases as the number of nodes increases. The largest jump between any two grid configurations occurs from a 2×4 grid to a 3×3 grid. Thus, a 3×3 grid is selected for the SOM analysis. While a higher spatial correlation can be achieved with larger number of grids, the 3x3 grid is able to capture the main variability pattern in the autumn Arctic sea ice at the sacrifice of some details that can be depicted by larger number of grids.

The frequency of occurrence for each SOM node is defined as the number of autumns that node represents divided by the total number of autumns (38) in the study period. For each SOM node, a linear trend is estimated by applying simple linear regression to its occurrence time series and the contribution of  that SOM pattern to the total 
[revised manuscript text omitted]

---

## Author Response (AR3)

Originally 'Contributions from intrinsic low-frequency climate variability to the accelerated decline in Arctic sea ice in recent decades'

Title changed to 'Changes in sea surface temperature and atmospheric circulation patterns associated with reductions in Arctic sea-ice cover in recent decades

- 5 Referee report:
  - General comments:

The authors have provided more information and revised the discussions in response to the referees' comments, and have improved their manuscript.

Some aspects of the SOM patterns are still not clear to me, and the discussion of mechanisms needs more detail to be clearer.

I am not convinced that asymmetries in North Pacific SST anomalies are as important as the authors claim, so that aspect could be more speculative.

Thus I recommend further revision as follows:

15 Specific comments:

30

Are composite results sensitive to the selection of the SOM matrix of patterns? As in the example in the supplement, patterns similar to node1 and node 9 also occur and dominate other choices of matrix, and the selection of years for composites is likewise similar for other array choices. The composite results are not sensitive to the selection of the SOM matrix of patterns.

- 20 On reading again the section on potential mechanisms, I think it could be made clearer and improved by providing more precise information about geographic regions mentioned, and linking more clearly to the sea ice concentration features in the nodes. e.g. line 162 (and 187): specify latitude/longitude for the region of interest in the western tropical Pacific; likewise line 168 for the 'region between two centers', line 171 for region of reduced sea ice concentration,
- 25 line 174 for the region of anomalous high pressure, line 176 for the region of anomalous low pressure, etc.

**Specific geographic locations have been added following the suggestion.**

Line 207/208: apart from opposite sign, the differences in SST patterns are rather small, and it is difficult to attribute the differences in high-latitude circulation to the SST patterns without more direct evidence, so this should be stated as a hypothesis. (The SST anomaly sign difference alone

- and nonlinear effects might be the main cause of any SST-related circulation differences.) It would be informative if one or two example scatterplots could be added to illustrate how particular regional values in ice cover and in atmospheric variables are related in in years making up composites.
- 35 This sentence has been revised, as suggested.

A scatter plot is produced (Fig. S1). During the years of Node 1 when sea ice concentration anomalies over Barents and Kara Seas (30°E-100°E, 70°N-85°N) are negative, the 500-hPa height anomalies in the region (0°-160°E, 60°N-90°N) are mostly positive. The opposite occurs during the years of Node 9.

45 Further minor comments:

It would be helpful to mention the use of composites in the abstract.

Added

line 25: '...remain a subject of ..'

Changed

50 line 31: '...On larger scales, ...'

Changed

line 50: '... (ENSO) events to the ...'

```
Added
```

55

line 78: what is the resolution of the OLR data grid?

The resolution of the OLR grid has been added.

Line 81: it would be helpful to add a statement about what is optimised in SOM Added

Line 85: I am not sure what the sentence 'All patterns in ...' means, as the patterns represent a subset of the input data.

60 Changed 'all patterns' to 'all vectors'

Line 94: it would be clearer to say 'The best matching SOM pattern for each autumn is determined on the basis of minimum Euclidean distance ....'. It would be useful to provide an equation for this. It would also be better to place this later in the paragraph (e.g. after line 105), after defining the patterns.

65 The sentence is modified, but we feel it is better to leave the sentence at where it is. Instead of adding an equation, a reference is added.

Line 107: It would be clearer to define frequency of occurrence as the number of autumns for which a pattern is selected as 'best matching' divided by the total number of autumns. Clarified 70 Line 110: I am not sure what is meant by 'linear regression of the number of the projections' : an equation would be useful.

Replace the word 'projections' by 'occurrences' and added an equation

Fig. 1: are the patterns normalised?

(It might be worth noting that the sign of the pattern is important, unlike EOFs for which the sign is arbitrary. An EOF analysis would give one pattern like nodes 1 and 9, with different signs of projection in early/late years in the data.)

The patterns were not normalized, which represent the realistic anomalous sea ice concentration.

Line 122: it would help to say that 24% corresponds to 9/38.

80 Added

75

90

Line 127: omit 'much' (the occurrence difference is between 5/38 and 3/38) Changed Line 130: ... Figs 2 and 3 Added

Fig. 4: are the labels correct? The SOM-associated trend seems larger than the observed trend. Also, the 'dots' indicating significance are very hard to see in the figure.

The label is right. The SOM-related trend is smaller than the observed trend. The dots are made less dense to make it easier to see.

Line 155: Suggest better to put 'To explore the processes associated with the spatial patterns ...' Changed

Line 213: better to also state the years used in the before and after 2000 composites. Added

Line 217: to me, the dominant feature of the SST patterns associated with node1 and node9 is their similarity, with small differences. Further evidence is required if circulation and sea-ice

95 concentration differences are to be attributed to the influence of the small SST pattern differences – perhaps you could state that further investigation is required to support your statements, particularly as this effect forms a large part of your conclusions in section 8. We added some discussions about the impact of SST.

Fig. 2 caption: Better to say that the figure indicates which mode best matched the observed sea ice concentration in each autumn.

Changed

100

Fig. 3 caption: Rather than 'explained', this figure just shows the linear trend associated with each pattern when projected on the observations?

Changed

105 Fig. 4: check the content, labelling and dots.

We revised fig. 4a.

Fig. 5 and Fig. 8: It would be helpful to repeat in the caption the actual years used for each composite.

Added

[revised manuscript text omitted]
 methods, which combines some features of clustering analysis and neural network, is more advantageous over traditional clustering and pattern extracting methods (Grotjahn et al., 2016).

- 195 The SOM technique is utilized to extract patterns of Arctic sea ice concentrations. A neural network-based method, SOM uses unsupervised learning to determine generalized patterns in complex data. The technique can reduce multidimensional data into two-dimensional array consisting of a matrix of nodes. Each node in the array has a reference vector that displays a spatial pattern of the input data. All patterns-vectors in the two-dimensional array represent the full continuum of states in the input data. The
- 200 SOM algorithm also is a clustering technique, but unlike other clustering technique, it does not need a priori decisions on data distribution. Unlike the EOF analysis, the SOM technique does not require the orthogonality of two spatial patterns. A detailed description of the SOM algorithm can be found in Kohonen (2001).

In this study, the SOM technique is used to categorize anomalous seasonal sea ice concentration patterns north of 50°N for autumn (September-October-November). Seasonal anomalies are calculated by subtracting the climatology for the season relative to the study period 1979-2016 from the seasonal means. The best matching SOM pattern to the sea ice anomaly patternThe\_anomalous sea ice pattern for each autumn is determined assigned to the best matching SOM pattern on the basis of minimum 210

calculated by the product of root-mean-square error and the square root of the number of grid points (Hewitson and Crane, 2002). Spatial correlations between sea ice field for each autumn and its corresponding best-matching SOM pattern are calculated and the overall mean spatial correlation coefficients obtained by averaging over the 38 autumns are used to determine the number of SOM nodes or grids to be used in the current analysis (Lee and Feldstein, 2013). Table 1 shows the overall mean

Euclidean distance, a measure of straight-line distance between two points in Euclidean space and

- 215 spatial correlation coefficients for several SOM grid configurations with nodes ranging from 2×2 to 4×5. As expected, the overall correlation increases as the number of nodes increases. The largest jump between any two grid configurations occurs from a 2×4 grid to a 3×3 grid. Thus, a 3×3 grid is selected for the SOM analysis. While a higher spatial correlation can be achieved with larger number of grids, the 3x3 grid is able to capture the main variability pattern in the autumn Arctic sea ice at the sacrifice of some
- 220 details that can be depicted by larger number of grids. The frequency of occurrence for each SOM node is defined as the number of autumns that node represents for which a pattern is selected as 'best matching' divided by the total number of autumns (38) in the study period. The contribution of each SOM pattern to the trends in the Arctic sea ice concentrations is calculated by the product of each SOM pattern and a rate determined by temporal linear regression of the number of the projectionsoccurrences of each pattern for each autumn (Johnson, 2013) represented by  $Y(y_1, y_2, \Box, y_i, \Box, y_n)$  onto the
- 225

time series of autumns between 1979 and 2016 represented by  $X(x_1, x_2, \Box, x_i, \Box, x_n)$  with n

equal to 38. The linear regression between Y and X is calculated using

 $y_i = \alpha + \beta x_i + \varepsilon_i$

Where  $\alpha$  is the intercept,  $\beta$  is the slope that we focus on, and  $\varepsilon_i$  is a residual term.

[revised manuscript text omitted]

**415**

- Gong, T., Feldstein, S., and Lee, S.: The Role of Downward Infrared Radiation in the Recent Arctic Winter Warming Trend. J. Climate, 30, 4937–4949, https://doi.org/10.1175/JCLI-D-16-0180.1, 2017.
- Grotjahn, R., Black, R., Leung, R., Wehner, M. F., Barlow, M., Bosilovich, M., Gershunov, A.,
   Gutowski Jr, W. J., Gyakum, J. R., Katz, R. W., Lee. Y.-Y., Lim, Y.-K., and Prabhat: North
   American extreme temperature events and related large scale meteorological patterns: a review
   of statistical methods, dynamics, modelling, and trends, Clim. Dyn., 46, 1151-1184, 2016,
   doi:10.1007/s00382-015-2638-6.

[revised manuscript text omitted]